# NUTS: Network for Unsupervised Telegraphic Summarization

## Abstract

Extractive summarization methods operate by ranking and selecting the sentences which best encapsulate the theme of a given document. They do not fare well in domains like fictional narratives where there is no central theme and core information is not encapsulated by a small set of sentences. For the purpose of reducing the size of the document while conveying the idea expressed by each sentence, we need more sentence specific methods. Telegraphic summarization, which selects short segments across several sentences, is better suited for such domains. Telegraphic summarization captures the plot better by retaining shorter versions of each sentence while not really concerning itself with grammatically linking these segments. In this paper, we propose an unsupervised deep learning network (NUTS) to generate telegraphic summaries. We use multiple encoder-decoder networks and learn to drop portions of the text that are inferable from the chosen segments. The model is agnostic to both sentence length and style. We demonstrate that the summaries produced by our model show significant quantitative and qualitative improvement over those produced by existing methods and baselines.

## 1 Introduction

Humans tend to relay information to others in a concise manner. Typically, we do not pass on information that we received in exactly the same format. More often than not, we tend to abridge and simplify it for the receiver. This captures the essence of summarization. Formally speaking, summarization refers to capturing all the information in a source, without compromising on understanding. There is a pressing need for summarization techniques to deal with the vast amount of textual data available nowadays.

In certain genres, like fictional narratives, each sentence might have an important role to play. This makes the domain of fictional narratives unsuitable for standard extractive summarization methods. Typically, extractive summarization works well if the source text revolves around a central theme with the same information reiterated across multiple sentences. For such text, picking a small number of relevant sentences is enough to summarize the text. Therefore, this method of extractive summarization is widely used for newswire articles (Lee et al., 2005), encyclopedic and scientific texts (Teufel & Moens, 2002) etc.

However, fictional stories and plays, for instance, do not always focus on a single theme. They describe a sequence of events and often contain dialogue. Information is not repeated and each sentence may contribute to developing the plot further. This problem is tackled in an alternate variation of extractive summarization known as telegraphic summarization. In this summarization technique, each sentence is considered an independent text source with each word acting as a segment. The telegraphic summaries for an input, read like a telegram, hence the name. For example, for an input sentence:*"An earthquake in Tokyo left 12 people dead"* the telegraphic summarization would be: *"earthquake Tokyo 12 dead"*. Recently, the advantages of telegraphic summarization over typical extractive techniques have been explored in Malireddy et al. (2018). The independent telegraphic summaries computed for each sentence in a document can be combined to form the summary of the whole document.

Current algorithms for telegraphic summarization rely on handcrafted rules (Grefenstette, 1998) or statistics based on syntax to infer important words in a sentence (Jing, 2000; Riezler et al., 2003;

Knight & Marcu, 2000). These methods, however, do not generalize for all cases. Additionally, the work by Lin (2003) suggests that pure syntax based methods cannot be used for general purpose summarization.

We overcome these limitations by exploring learning based approaches for telegraphic summarization. We propose an unsupervised deep network which implicitly learns to retain significant words in a sentence. We try to reconstruct the source text (which is the input to our network), from an abridged version of it generated at an intermediate stage. This abridged version can be used as the telegraphic summary of the source. Since the proposed architecture follows an unsupervised approach, we do not need any labeled text-summary pairs for our method. Another advantage of our algorithm is that it is agnostic to variations in genre, sentence length, vocabulary, language etc.

## 2 RELATED WORK

Early work on text summarization started around the mid $20^{th}$ century. In one of the earliest work by Luhn (1958), summarization was used to generate abstracts automatically from the excerpts of technical papers and magazine articles. Since then, various algorithms for text summarization have been proposed for multiple domains like newspaper articles (Wu & Hu, 2018), scientific articles (Teufel & Moens, 2002) and blogs (Hu et al., 2007).

A comparative study in Ceylan et al. (2010) revealed that a single text summarization algorithm does not perform equally well for all domains. This is because of differences in the nature and style of the sources. Therefore it was concluded that separate algorithms need to be designed for different domains of text sources.

Traditionally, most text summarization algorithms have focused on summarizing newspaper and scientific articles (Luhn, 1958; Teufel & Moens, 2002; Lee et al., 2005). Although, there have been recent attempts to summarize fictional stories (Kazantseva & Szpakowicz, 2010; Lloret & Palomar, 2009; Mihalcea & Ceylan, 2007), the objective of these algorithms is to let the reader decide if the text is interesting enough to read. Therefore, these algorithms do not intend to encapsulate the entire information of the source text.

An early general purpose text summarization algorithm for telegraphic purposes was proposed in Grefenstette (1998). This was a rule-based algorithm, with each rule allowing some specific components of the source sentence in the summary. The first (and the most drastic) generated a stream of all the proper nouns in the text. The second generated all nouns present in the subject or object position. The third, in addition, included the head verbs. The least drastic reduction generated all subjects, head verbs, objects, subclauses, prepositions and dependent noun heads. Since this algorithm is based on definitive guidelines, it does not generalize well for shorter sentences. Additionally, it is not clear which rule should be applied to a sentence to generate its appropriate summary.

There have also been numerous approaches using statistics based on syntax to generate telegraphic summaries (Jing, 2000; Riezler et al., 2003; Knight & Marcu, 2000). Typically, these approaches drop words from the source which are not related to the neighboring words, whilst trying to maintain the grammaticality of the summary. The grammatical correctness of the summary is checked using statistical models. These approaches do not tend to give accurate results for shorter inputs as the constraints for the removal of words are difficult to satisfy.

Recently, deep learning based methods have been enormously successful in the domain of feature learning with little or no manual intervention. These approaches have been applied to core Linguistics problems such as machine translation (Bahdanau et al., 2014), language modeling (Mikolov et al., 2010) and image captioning (Johnson et al., 2016). Although these approaches achieve better results than handcrafted algorithms, they are supervised approaches which require a huge amount of data. Attempts have been made to counter this by exploring unsupervised learning-based algorithms, eliminating the need for labeled ground truth data.

In this work, we have proposed a new architecture for telegraphic summarization, known as NUTS. The proposed architecture implicitly learns to retain the significant segments of the source through several encoder-decoder networks (Sutskever et al., 2014) to generate its telegraphic summary.

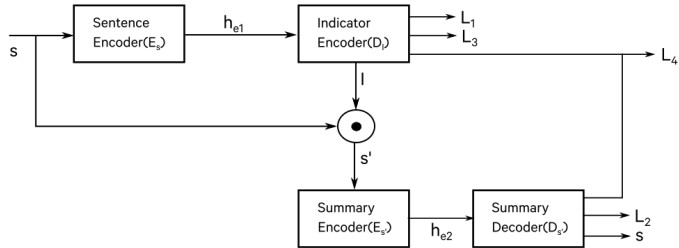

Figure 1: The figure shows the proposed NUTS architecture model. The information in a given input sentence **s** is encoded into a vector $h_{e1}$ as the final hidden state of Encoder $E_s$. This hidden state vector $h_{e1}$ initializes decoder $D_I$, which generates an indicator vector **I**. The element-wise multiplication of **I** and **s** is used to construct the masked sentence **s'**. This masked sentence is encoded into a vector($h_{e2}$) using $E_{s'}$. This vector is then used to initialize decoder $D_{s'}$ which aims to reconstruct **s** from the masked sentence.

## 3 NUTS MODEL DETAILS

In this section, we explain the details of the proposed model. We first describe the building blocks of the architecture. Thereafter, we introduce the various components of the loss function and explain each loss term and its usage.

### 3.1 ARCHITECTURE DETAILS

Given an input sentence **s** = $w_1$, $w_2$ ..., $w_k$ containing $k$ words, the network tries to reconstruct the same sentence **s**, with fewer number of words. The network architecture is such that it drops the words it deems inferable for the final reconstruction. The network generates an indicator vector **I** = $I_1$, $I_2$ ..., $I_i$, ..., $I_k$ at an intermediate stage. The value of this indicator vector corresponds to the presence of word $w_i$ in the summary subset **T** of the sentence set **S**. The words present in **T** are used as the telegraphic summary of the sentence in order of their occurrence in **s**.

Formally, the network tries to find an $I^*$ such that the probability $p(s|s \odot \mathbf{I})$ is maximized and $\sum_{t=1}^{k} I_t$ is minimized, jointly. The probability $p(s|s \odot I)$ can be decomposed further as shown in Equation 1

$$I^* = \arg\max_I \prod_{t=1}^{k} p(w_t|(w_1 \times I_1), (w_2 \times I_2), ..., (w_{k-1} \times I_{k-1})) \tag{1}$$

The entire architecture is built using RNN with LSTM cells proposed in Hochreiter & Schmidhuber (1997). The equations for LSTM gates are mentioned in Equation A.1 in the Appendix section.

The proposed architecture has the following major components (illustrated in Fig. 1):

1. **Sentence encoder** ($E_s$) encodes the input sentence **s**. Every word $w_i$ in the sentence is converted into a $d$-dimensional vector $e_i$. The sequence of these embedded vectors is fed as input to the encoder at each time step. The final hidden state of $E_s(h_{e1})$ acts as a sentence embedding for **s**.

2. **Indicator decoder** ($D_I$) is initialized by the final hidden state of $E_s$. The output of this decoder at each time step is passed through a network of two fully connected layers to generate a single output value. We intend this output value to be close to either zero or one, denoting the value of indicator vector **I**.

3. **Summary encoder** ($E_{s'}$) encodes the masked sentence $s'$ = **s** $\odot$ **I**, where $\odot$ represents element-wise multiplication. Therefore, the words corresponding to $\mathbf{I}_i \approx 0$ are effectively skipped. The final hidden state of $E_{s'}$ acts as a sentence embedding for **s'**.

4. **Summary decoder** ($D_{s'}$) is initialized by the final hidden state of $E_{s'}(h_{e2})$. This decoder aims to regenerate the input sentence **s** from **s'**. This motivates $D_I$ to generate **I** in such a way that **s** can be easily reconstructed from **s'**.

The output at each time step from the LSTM cell in $D_{s'}$ are fed to a dense layer, $W_s$. This dense layer computes the distribution over the vocabulary words from the decoders' hidden states.

## 3.2 Loss function

In order to understand the function of each of the modules of the architecture, we outline the need for each module's loss term and their effects on the output of the network.

1. **Summary length loss** ($L_1$): ensures that the ratio between the lengths of $\mathbf{s'}$ and $\mathbf{s}$ is close to a specific value $r$. This loss is calculated from the output of $D_I$. We have observed that extreme values of $r$ fail to generate the desired output. In our experiments, low values of $r$ hindered the ability of the summary decoder $D_{s'}$ to reconstruct $\mathbf{s}$ from $\mathbf{s'}$. On the other hand, if r is very large, no words are dropped and $\mathbf{s'}$ and $\mathbf{s}$ become equal. Therefore, the value of r is set such that the summary is neither too short nor too long. Mathematically, $L_1$ can be represented as shown in Equation 2. We calculate the value of $Len(s')$ in this equation as the sum of elements of $\mathbf{I}$. The optimum value of r for our experiments was found to be 0.65.

$$L_1 = \left( \frac{Len(s')}{Len(s)} - r \right)^2 \tag{2}$$

2. **Summary Decoder Reconstruction loss** ($L_2$) ensures that the words retained in $\mathbf{s'}$ are sufficient for reconstructing $\mathbf{s}$. The $L_2$ loss is calculated from the output of $D_{s'}$. It aims to maximize the probability of occurrence of $w_i$(the $i^{th}$ word in $\mathbf{s}$), given the final hidden state of $E_{s'}$ ($h_{e2}$) and all previous words in the masked sequence($w'_{<i}$), encapsulated by the hidden state vector from the previous time step. The mathematical representation of $L_2$ is shown in Equation 3.

$$L_2 = - \sum_{i=1}^{Len(s)} logP(w_i | w'_{<i}, h_{e2}) \tag{3}$$

3. **Binarization loss** ($L_3$) is calculated from the output of $D_I$. Ideally, we want all values in $\mathbf{I}$ to be either zero or one. However, if the target indicator is set to these hard values, non-differentiability is introduced into the network. Therefore, we relax the requirement of hard values and fix the range of $D_I$ output as $[0,1]$ using the sigmoid activation function. Additionally, we model $L_3$ such that the outputs tend to be distant from the mid-value of 0.5. This pushes the outputs close to zero or one, effectively fulfilling the objective of $\mathbf{I}$. Mathematically, this is achieved using Equation 4. A larger value of $b$ results in a higher penalty for mid-ranged values of $\mathbf{I_i}$. The value of $a$ is such that $L_3$ is always non-negative. In our experiments, the best value of $b$ was found to be 5.

$$L_3 = \frac{\sum_{i=1}^{Len(s)}(a - b(I_i - 0.5)^2)}{Len(s)} \tag{4}$$

4. **Linkage loss** ($L_4$) In our experiments we noticed that without an additional loss to govern how the words are masked, the network tends to learn repetitive masking patterns. Such skewed patterns are undesirable - while certain positions of $\mathbf{s}$ are well represented in $\mathbf{s'}$, other positions are entirely missing. A few examples of such summaries are shown in Fig. A.1 of Appendix 1. This motivates the need to couple ease of decoding with words dropped from the summary. We introduce a novel 'linkage' loss function to achieve this.

   The linkage loss facilitates the network to drop words which are deemed inferable. It is applied to the outputs of $D_I$ and $D_{s'}$ simultaneously. Therefore, it correlates the indicator vector and its effect on reconstruction. It penalizes the network if a) it decides to mask a word but is unable to reconstruct it later or b) it decides to include a word which it could decode easily. The mathematical equation describing the linkage loss is shown in Equation 5. The variable $\chi_i$ denotes the relative ease of decoding word $w_i$, given $w'_{<i}$ and $h_{e2}$. The value of $\chi_i$ lies between 0 and 1 (both included). A larger value $\chi_i$ indicates that $w_i$ is difficult to decode. The value of $\chi_i$ is calculated using Equation 6. The addition of linkage loss also results in significantly better results significantly as seen in Section 5.

$$L_4 = \sum_{i=1}^{Len(s)} \left( \mathbf{I_i}e^{(1-\chi_i)} + (1 - \mathbf{I_i})e^{\chi_i} - 1 \right) \tag{5}$$

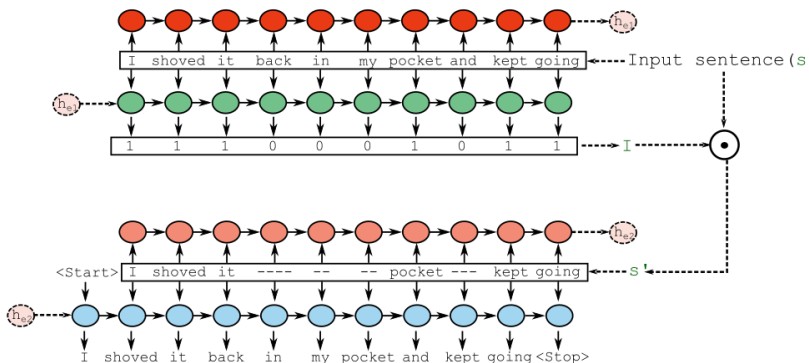

Figure 2: The figure shows a detailed flow of the NUTS architecture with an example input.

$$\chi_i = \frac{|logP(w_i|w'_{<i}, h_{e2})|}{\max_{1 \leq i \leq Len(s)}|logP(w_i|w_{<i}, h_{e2})|} \tag{6}$$

It can be seen from Equation 5 that $L_5$ is minimized when either a) $\chi_i = 0$ and $\mathbf{I}_i = 0$ simultaneously (signifying that $w_i$ is easy to decode and should be dropped) or b) $\chi_i = 1$ and $\mathbf{I}_i = 1$ simultaneously (indicating that hard-to-decode words should be retained).

The cumulative loss function $(L)$ is a linear combination of the losses described above. Mathematically, the equation for L is represented in Equation 7.

$$L = \lambda_1 L_1 + \lambda_2 L_2 + \lambda_3 L_3 + \lambda_4 L_4 \tag{7}$$

The weights $\lambda_1$, $\lambda_2$, $\lambda_3$, and $\lambda_4$ have been set to 3, 2, 50 and 3 respectively for our experiments. Since this is an unsupervised approach currently the weights are experimentally determined and can be fine-tuned on supervision.

### 3.3 WEIGHTED LINKAGE

We observed that some words are an integral part of the sentence and should never be dropped. Their absence from the summary can entirely change the meaning of the summary/make the summary meaningless (e.g. the word **not** in source sentence *'He is not my friend'*). However, the Summary Decoder $D_{s'}$ could infer some of these words despite their absence from the masked sentence. We found that mostly subject and negation words fall into this category.

Therefore, to make the summary more aligned with the meaning of the source text, we provide an additional 'retention' weight $(\omega_i)$ for each word along with sentence **s** in the input. We use these weights to highlight the importance of retaining these words. This is achieved by modifying the variable $\chi_i$ in the linkage loss described in Equation 5. The modified variable $(\chi'_i)$ can be obtained using the relation shown in Equation 8. The need for weighted linkage loss is demonstrated with an example in Fig. A.2 of Appendix 1. Although the quantitative scores are not affected significantly, we achieve better qualitative results in terms of retention of meaning due to addition of weighted linkage.

$$\chi'_i = \frac{\omega_i \times |logP(w_i|w'_{<i}, h_{e2})|}{\max_{1 \leq i \leq len(s)}(\omega_i \times |logP(w_i|w_{<i}, h_{e2})|)} \tag{8}$$

It should be noted that the weight input is only required at training time and no additional input other than the sentence is required once the model has been trained.

A detailed flow of the network with an example input is provided in Fig. 2.

## 4 EXPERIMENTS

We have used the BookCorpus dataset introduced in Zhu et al. (2015) for training the model. This dataset contains 11038 books in 16 different genres (e.g. romance, comedy, science fiction, teen

etc.). The total number of sentences and distinct words in the dataset are 74,004,228 and 1,316,420 respectively. The dataset is widely varied in terms of narratives, emotions and text style.

We created a dataset of 500 sentences for test purposes. These sentences were collected from fictions of several genres. This dataset is similar to the BookCorpus dataset used for training. The sentence length is varied between 5 and 30, with an equal number of examples for each length. The sentences are annotated using the guidelines stated below:

1. A word is the smallest unit in the sentence. It should *not* be broken further. ("waiting" $\not\to$ "wait").

2. The order of occurrence of words should be preserved in the summary.

3. The summary should be minimal without changing the meaning of the source.

4. The first occurrence of subject words should be included. Thereafter, subject words and corresponding pronouns should only be included if necessary.

Five different users (all fluent in English) summarized 100 different sentences each. We verified the consistency of the reference summaries using the inter-annotator agreement Kappa score. All users were asked to cross-annotate additional 20 sentences for this purpose. The users had an agreement Kappa score of 0.82, thus ensuring the gold standard of the summaries. The average relative length of the summaries, created using these guidelines, with respect to the source sentence was 0.702. Some examples of the summaries generated using these guidelines are shown in Table A.1 in Appendix 1.

## 4.1 DETAILS OF TRAINING

We train the model on the entire BookCorpus dataset. All sentences were divided into buckets based on the number of words. We considered only those buckets where the length was between 5 and 30 words in our experiments. All other sentences were discarded. This was done to ensure that the input was neither too short nor too long. We consider an equal number of sentences from each of the 26 buckets, to make the network agnostic to sentence length. The number of sentences in the smallest bucket was 375,000. Thus, a total of 9,750,000 sentences were considered from the original dataset. 10% of these sentences were used as the validation set and the rest for training. The sentences are input to the network in mini-batches of size 128.

We restricted our vocabulary to 20000 most frequent words from the dataset (Kiros et al., 2015). Each word was embedded as a 300-dimensional vector. All the encoders and decoders are modeled using RNNs with LSTM cells of size 600. The weights of all the RNNs were initialized normally $\mathcal{N}(\mu = 0, \sigma = 0.1)$.

The output from $D_I$ is passed through two fully-connected layers to form **I**. The hidden layer has 150 units with ReLU activation and the output layer has a sigmoid activation. The output of $D_{s'}$ at each time-step is multiplied by a matrix $W_s$, thereby projecting its output from 600-dimensional space to 20000-dimensional vocabulary-space. Softmax activation is then applied over the output of the dense layer to compute probability distribution over the vocabulary. We used Adam (Kingma & Ba, 2014) as the optimization algorithm, with the initial learning rate set to 0.001, $\beta_1$=0.9 and $\beta_2$=0.999. Gradients were clipped when they became larger than 1.0.

## 5 RESULTS

We compare the NUTS model with the two baseline methods (explained below), and a rule-based algorithm proposed by Grefenstette (1998), due to the absence of any recent algorithm to generate telegraphic summaries. The Grefenstette algorithm provides eight levels of summaries. Each level allows for varying amount of information to be retained at the cost of summary length. In our experiments, we consider the summary generated at the last level (most informative) as the output.

## 5.1 BASELINE

We use the stop words' list as a baseline indicator of the words which do not contribute to the summary of a sentence. Therefore, for any given input sentence, the telegraphic summary is formed

by removing all the stop words present in the input. The experiments are conducted using the top $1\%$ (200) of the total number of words in the vocabulary. The summary generated after removing the stop words from this list is referred to as **B1**.

We also use an alternate baseline motivated from the TextRank algorithm (Mihalcea & Tarau, 2004). A graph is constructed with the input words as nodes. The weight of the edge between two words, w1 and w2, is equal to the cosine similarity between the corresponding word vectors, e1 and e2. Pre-trained Glove vectors (Pennington et al., 2014) are used to compute e1 and e2. We then run the PageRank algorithm (Page et al., 1998) till convergence and select the top nodes as the summary. Hereafter, this baseline is referred to as **B2**.

## 5.2 QUANTITATIVE EVALUATION

Traditionally, the evaluation of summarization algorithms for a document is done using ROUGE as proposed in Lin (2004). ROUGE-N evaluation is based on n-gram co-occurrence between the system and reference summaries. The equation used to calculate ROUGE-N scores can be seen in Equation 9, where N stands for the length of the n-gram, $gram_n$ and $Count_{match}(gram_n)$ is the maximum number of n-grams co-occurring in a candidate summary and a set of reference summaries. In our experiments, each sentence is considered to be an independent document. The reference summaries are generated using the guidelines stated in Section 4.1. All system generated summaries are evaluated against the reference summaries for four values of N(1,2,3,4).

$$ROUGE_N = \frac{\sum_{s \in Reference\ summaries} \sum_{gram_n \in S} count_{match}(gram_n)}{\sum_{s \in Reference\ summaries} \sum_{gram_n \in S} count(gram_n)} \tag{9}$$

Additionally, we also thought that it would be interesting to compare various algorithms based on their summary lengths. We measured the length of the system generated summaries against the input sentence and the reference summaries for this purpose. These metrics are referred to as summary factor($s_f$) and length factor($l_f$) respectively. The scores are calculated using the formulae shown in Equation 10 and Equation 11.

$$s_f = \frac{Len(summary_{system})}{Len(\mathbf{s})} \tag{10}$$

$$l_f = \frac{Len(summary_{system})}{Len(summary_{reference})} \tag{11}$$

The ROUGE-N, $s_f$ and $l_f$ scores are calculated on the entire test dataset of 500 sentences. The mean scores for each algorithm are presented in Table 1. It can be clearly seen that the proposed architecture outperforms the existing algorithm and the baseline methods by a significant margin. The NUTS architecture increases the average ROUGE-N score by $0.164$ from the next best-performing algorithm. Additionally, the average summary length is also close to that of the reference summaries as can be seen by the $l_f$ scores. Although the $s_f$ scores suggest that the summaries generated using **B1** are most concise, the scores are less because the summaries hardly retain any words. This can be verified by observing the other metrics used for comparison. The scores demonstrate that NUTS implicitly balances summary length and content retention, i.e. it learns to retain most words in shorter sentences while shortening the longer sentences simultaneously. There is no other recent algorithm attempting to solve the telegraphic summarization, especially involving learning, to the best of our knowledge. A primary reason for this could be lack of annotated data, which is also the motivation behind the unsupervised approach proposed in this work.

Table 1: Comparison of various algorithms on test dataset.

|  | N=1 | N=2 | N=3 | N=4 | $s_f$ | $l_f$ |
|---|---|---|---|---|---|---|
| B1 | 0.665 | 0.337 | 0.186 | 0.10 | **0.389** | 0.559 |
| B2 | 0.707 | 0.346 | 0.156 | 0.074 | 0.611 | 0.878 |
| Grefenstette | 0.672 | 0.279 | 0.082 | 0.030 | 0.644 | 0.920 |
| **NUTS** without $L_4$ | 0.715 | 0.378 | 0.172 | 0.069 | 0.784 | 1.12 |
| **NUTS** | **0.816** | **0.545** | **0.372** | **0.263** | 0.689 | **0.983** |

### 5.3 Qualitative evaluation

The ROUGE-N and length factor scores demonstrate the content adequacy and relative sentence length similarity between the system and reference summaries. However, these scores can not comment on the readability of the system generated summaries. The ROUGE-N scores only measure the overlap between the reference and system summary and do not account for the coherence of the summary with the original sentence. Some examples demonstrating these shortcomings of the ROUGE-N scores can be seen in Appendix 1 (Fig. A.3).

We conducted a study to validate the performance of various systems in terms of coherence and meaning between the sentence and its summary. We selected 10 English speaking participants to undertake the study. Each participant was provided with 20 input sentences and four summaries for each sentence. They were asked to choose the summary which best captured the gist of the sentence. Among the 20 sentences, five sentence-summary pairs were present for a sanity check. One of the summaries in each of these five sentences was manually annotated, which the participants were expected to choose. All the choices of the participants who chose a different summary more than once out of these five sentences were discarded from evaluation. As a result, the choices of two participants were not considered. Thus, the remaining 8 participants and their choices for the other 15 questions were examined. Therefore, a total of 120 responses were finally considered for evaluation.

The choices for the remaining sentences corresponded to the four system generated summaries. Out of the 120 responses considered, the participants selected summary generated by the NUTS model 104 times(86.67%). Fig. A.4 in Appendix 1 shows examples of some sentence-summary pairs along with their references. As can be seen, the NUTS model mostly generates summaries close/equal to the reference summaries(sentence 2). We also preserve the meaning of the input sentence (the retention of 'not' in the third sentence and 'own' in the fourth sentence). It is noteworthy that the nuances are captured despite significant shortening of the input.

The figure also shows some failure cases of the model. The words 'origami' and 'thrashers' may have been masked in sentence 5 and 6 respectively due to the limited vocabulary of the encoder. This may be tackled by expanding the model's vocabulary as suggested in Kiros et al. (2015).

## 6 Future Work

We plan to extend the work proposed in this paper to further reduce the summary length corresponding to a source text. We propose to stack the telegraphic outputs of multiple sentences as a new input to our algorithm and iteratively reduce the summary length across multiple sentences. Alternatively, the addition of context might help to generate concise telegraphic summaries for the entire text.

Another active area of our work is to use sentences selected by existing extractive summarization techniques (Wu & Hu, 2018) as our input. This can enable these algorithms to extract additional sentences in their output. We also plan to develop an algorithm to convert telegraphic inputs into fully grammatical sentences.

## 7 Conclusion

We present a learning-based technique for the task of telegraphic summarization. This is the first attempt to use a deep learning for telegraphic summarization. Furthermore, the proposed network is completely unsupervised and removes the need for labeled summary texts. The network implicitly learns the trade-off between readability and understanding, with respect to the source text. We demonstrate through experiments that our network substantially improves results over the baselines. Additionally, we also conducted qualitative experiments and demonstrate that the proposed network generalizes over length and genre of the source text.

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

## A    APPENDIX 1 : ADDITIONAL DETAILS AND EXAMPLES

**LSTM Details**

LSTM refers to Long Short-Term Memory, enabling the network to retain memory across time-steps. Additionally, LSTM is designed such that the error gets backpropagated across time-steps efficiently. The equations for various gates and the outputs of a LSTM are presented below.

$$
\begin{aligned}
i_t &= \sigma(W_i d_t + U_i h_{t-1} + b_i) \\
f_t &= \sigma(W_f d_t + U_f h_{t-1} + b_f) \\
o_t &= \sigma(W_o d_t + U_o h_{t-1} + b_o) \\
\hat{c}_t &= \tanh(W_c d_t + U_c h_{t-1} + b_c) \\
c_t &= i_t \odot \hat{c}_t + f_t \odot c_{t-1} \\
h_t &= o_t \odot \tanh(c_t)
\end{aligned}
\tag{A.1}
$$

where, $d_t$ is the value of an input to the cell at time $t$; $h_t$ is the value of the hidden state at time $t$; $W_i$, $W_f$, $W_o$, $W_c$ are the weight matrices corresponding to $d_t$; $U_i$, $U_f$, $U_o$, $U_c$ are the weight matrices

corresponding to $h_{t-1}$; $b_i$, $b_f$, $b_o$, $b_c$ are the bias values for each gate. The output of each gate at time $t$ is represented as $i_t$, $f_t$, $o_t$, and $\hat{c}_t$. The output of the LSTM cell at time t is represented by $c_t$. The operator $\odot$ refers to element-wise multiplication between corresponding operands.

**Additional Examples**

In this sub-section, we illustrate the effects of linkage loss and weighted linkage through examples. Additionally, we also show some examples of the sentence-summary pairs from the test dataset we have generated. Finally, we demonstrate the effectiveness of NUTS architecture to retain the meaning of the input sentence and reduce the summary length simultaneously.

```
Input       : Then I headed back downstairs to help set the table.
Reference   : I headed downstairs help set table.
Pattern 1   : ---- - ------ ---- downstairs to help set the table.
Pattern 2   : ---- I headed ---- downstairs to ---- set the -----.
```

Figure A.1: The figure shows examples of undesirable patterns learned by the network in the absence of linkage loss. In Pattern 1, the model masks the entire first half. Similarly, in the Pattern 2, the model learns to mask every third word. Even though the resultant summary length for both patterns is equal to that of the reference summary, neither of them are able to capture the semantics of the input.

```
Input           : I did not have much time.
With weights    : I not have time.
Without weights : have time.
```

Figure A.2: The figure demonstrates the effects of weighted linkage. When the retention weights are not taken into account, the subject and negation words are masked. However, this is rectified when retention weights are introduced, thereby preserving the meaning of the sentence.

Table A.1: Examples of sentence and summary pairs used for testing the network.

| Sentence | Summary |
|---|---|
| I shoved it back in my pocket and kept going | I shoved it pocket kept going |
| I kept my eyes on the shadowed road watching my every step | I kept eyes on road watching every step |
| I thumbed the keypad and opened the message Seth had sent me | I thumbed keypad opened message Seth sent |

```
Input sentence   : Delhi is not the place where you want to be after dark.
Reference summary: Delhi not place you want be after dark.
System summary   : Delhi place you want be after dark.
ROUGE-1 score    : 0.875
```
(a)

```
Input sentence   : I would like to get home before my dad wakes her up.
Reference summary: I like get home before dad wakes her.
System summary   : I like get home before dad wakes.
ROUGE-1 score    : 0.875
```
(b)

Figure A.3: The figure illustrates the inability of ROUGE-N scores to capture the difference in meaning between system and reference summaries. In (a) the ROUGE-N score reports a similarity of 0.875. However, the absence of one word has completely changed the meaning of the summary. Similarly in (b), the one word which is missing is non-inferable and makes the summary open to multiple interpretations, despite a high ROUGE-N score with the reference summary.

```
Input sentence        : Two bank robbers were shot dead last Friday when a high-speed car
                        chase ended a gun battle in the city center.
Reference summary     : Two robbers shot dead Friday high-speed car chase ended gun
                        battle city-center.
B1 summary            : bank robbers shot dead Friday high-speed chase ended gun
                        battle city center.
B2 summary            : Two bank robbers were shot friday car ended in gun the city center
Grefenstette summary  : robbers shot last chase ended in battle in center.
NUTS summary          : Two bank robbers shot dead last friday when car chase ended gun
                        battle city center.

Input sentence        : The two men inside were reported dead on arrival in hospital.
Reference summary     : two men inside reported dead arrival hospital.
B1 summary            : The men inside reported dead arrival hospital.
B2 summary            : men inside were reported dead hospital.
Grefenstette summary  : men reported on arrival in hospital.
NUTS summary          : two men inside reported dead arrival hospital.

Input sentence        : We commonly do not remember that it is, after all,
                        always the first person that is speaking.
Reference summary     : We do not remember it is always first person speaking.
B1 summary            : We commonly remember is, all, always person speaking.
B2 summary            : We commonly not that it is after all always the.
Grefenstette summary  : We not remember it is person speaking.
NUTS summary          : We commonly not remember it after all always first person is speaking.

Input sentence        : I looked at my own shadow that danced behind me.
Reference summary     : I looked own shadow danced behind.
B1 summary            : shadow danced.
B2 summary            : I looked at own shadow that.
Grefenstette summary  : I looked at shadow that danced behind me.
NUTS summary          : I looked own shadow danced behind.

Input sentence        : Or maybe I could finish a painting tonight and watch mom do origami.
Reference summary     : maybe I finish painting tonight watch mom origami.
B1 summary            : maybe finish painting tonight watch mom origami
B2 summary            : Or maybe I finish tonight watch mom.
Grefenstette summary  : I finish painting tonight watch mom origami.
NUTS summary          : Or maybe I finish painting tonight watch mom.

Input sentence        : Thrashers were probably one of the most feral and nasty monsters
                        I've ever encountered.
Reference summary     : Thrashers probably most feral nasty monsters I've encountered.
B1 summary            : Thrashers probably feral nasty monsters I've ever encountered.
B2 summary            : Thrashers were probably feral nasty I've ever encountered.
Grefenstette summary  : Thrashers were monsters I've encountered.
NUTS summary          : were probably one most feral nasty monsters I've ever encountered.
```

Figure A.4: The figure shows some examples of system generated summaries for a given sentence along with their reference summaries.

