# OpenReview forum: "NUTS: Network for Unsupervised Telegraphic Summarization"
_ICLR.cc/2019/Conference_

### Official Review · AnonReviewer2 · 2018-10-29
**Novelty and advantage of the proposed methods are limited**

**Rating:** 4
**Confidence:** 4

**Review:**

The paper explores unsupervised deep learning model for extractive telegraphic summaries, which extracts text fragments (e.g., fragments of a sentence) as summaries. The paper is in general well structured and is easy to follow. However, I think the submission does not have enough content to be accepted to the conference.

First, in term of methodology (as described in Section 3), the paper has little novelty. There has been intensive study using various deep learning models on summarization. The models described in the paper contain little novelty compared with previous work using autoencoder and LSTM for both extractive and abstractive summarization.

Second, the paper claims contributions on using deep learning models on telegraphic summarization, but the advantage is not well demonstrated. For example, the advantage of the resulting summary is not compared with state-of-the-art sentence compression models with intrinsic evaluation or (probably better) with extrinsic evaluation. (By the way, it is interesting that the paper argues the advantage of using telegraphic summaries for fictional stories but actually gives an example which looks also very typical in news articles (the “earthquake Tokyo 12 dead” example).)

Third, there has been much work on speech summarization that summarizes with the “telegraphic” style (this is natural, considering speech transcripts are often non-grammatical, and “telegraphic” style summaries focusing on choosing informative fragments actually result in usable summaries.) The author(s) may consider discussing such work and compare the proposed methods to it.

---

> ### Author Response · Authors · 2018-11-25
> **Response to Reviewer 2**
>
> R2 : Concerns about novelty of the work
>
> While LSTMs and autoencoders find several applications in the domain of summarization and NLP in general, the model (composed of LSTMs) proposed in this paper is in fact novel. The architecture is intuitively motivated to solve the problem at hand and the objective for each loss function is clearly explained and finds a parallel in the problem statement. Linkage loss, especially, is a novel contribution by this paper as it  intuitively ties summary length with ease of inferability.
>
> R2 : Comparison with state-of-the-art and issue with illustrative example
> The paper does lack in making comparisons to the state-of-the-art sentence compression models and we’d like to thank R2 for pointing this out. But the example R2 points out is one among several example sentences presented in the paper. This example was simply used to demonstrate what telegraphic summaries look like. Telegraphic summarization itself can be applied to various domains such as fiction, research papers and even news articles, hence the example. The paper simply states that the benefits of telegraphic summarization are most clearly seen in the domain of fiction.
>
> R2 : Comparison with speech summarization
> Speech summarization does lead to telegraphic summaries (due to the nature of the source text itself), but such methods are prone to use acoustic signals etc as features therefore making a direct comparison infeasible. However, the domain of speech transcripts could be an interesting application for our method and we’d like to thank R2 for pointing this out.

---

### Official Review · AnonReviewer3 · 2018-11-02
**The proposed linkage loss for telegraphic sentence compression is inspired, but some choices still seem arbitrary and the usefulness of the task needs to be better motivated.**

**Rating:** 4
**Confidence:** 4

**Review:**

The authors consider the problem of telegraphic sentence compression: they train a system in an unsupervised fashion to predict which words can be dropped from a sentence without drastic loss of information. To that end, they propose a new auto-encoding type architecture which uses the extracted words as latent code, and, most importantly, a linkage loss which relates a word's perplexity given the summary of its left context to its likelihood of being retained. The model itself is sober and well motivated, and the linkage loss is, to the best of my knowledge, original. The authors show that their method outperforms some simple baselines in terms of ROUGE and compression on a small human-annotated test set.

The paper is generally well written, although the initial presentation of the model could be made a little clearer (it is not obvious from the text that the Decoder takes the text as input -- Figure 2 helps, but comes a couple pages later). However, the authors fail to appropriately justify the choice of their hyper-parameters (e.g. "The optimum value of r for our experiments was found to be 0.65", "the best value of b was found to be 5", "The weights λ1, λ2, λ3, and λ4 have been set to 3, 2, 50 and 3 respectively for our experiments" -> how is "best" measured on the validation set, which does not have gold references?). The choice of the specific sparsity constraint (one could as well imagine using a simpe L1 regularization for the Binarization loss) and of \Chi_i (why not simply use the likelihood?) could also be better motivated.

The model also relies on a hand-crafted rules (Section 3.3) whose effect needs to be made more evident. What weights are used in practice? How were they chosen ("We observed that..." needs to be further developed)? The authors claim that "the quantitative scores are not affected significantly", but that is presumably only the ROUGE score, what about annotator's preferences?

Most importantly, however, the task of telegraphic sentence compression, whose usefulness is not a priori obvious, is barely motivated.  The author refer to "Malireddy et al. (2018)" for a justification, but it is important to note that the latter provides a telegraphic summary of a whole document, with a compression factor of 0.37. The claim is that the concatenation of the telegraphic sentence compression can act as a summary of a whole document, but given the fact that compression for individual sentences is closer to 0.69, this is yet to be demonstrated. And even if that were true, it is unclear whether the cognitive load of reading a sequence of telegraphic sentences would be that much lower than that of reading the original text.

This paper presents some interesting ideas and is well written, but the content is not quite sufficient for publication. In addition to the clarifications and justifications requested above, the authors are encouraged to apply there methods to full lengths documents, which would make for a more substantial contribution.

---

> ### Author Response · Authors · 2018-11-25
> **Response to Reviewer 3**
>
> R3 : Regarding the setting if hyperparameters in the loss functions and weights assigned to each model
>
> The hyperparameters are set by empirically observing the loss function patterns and making sure that the entire loss value is not driven by a single loss, and the decrease is due to a combination and trade-off between all individual losses. For example we observed that high values of λ1 would cause the summary length loss to decrease rapidly, reducing the summary to one or two words and the model would not recover after that. We experimented on the hyperparameters till no such effects were observable for multiple runs. We agree that the hyperparameters can further be fine-tuned.
>
> R3 : The choice of sparsity constant and specific expression for \Chi_i
>
> The particular sparsity constraint was chosen to ensure that the output of the indicator decoder, for all practical purposes, always contains values equal to zero or one, and not just a tendency for sparse values, as is assisted by L1 regularization. Additionally, \Chi_i is a variant of log likelihood, which, in addition to giving a measure of how hard each word is to infer, also normalizes the values across each word in the sentence. These normalized values were seen to be a better measure of inferability. But we agree that the above explanation can be present in the paper itself to better motivate the choice of loss functions.
>
> R3 : Handcrafted setting of words to be retained and weight settings
>
> The selection of negation and subject words is done using POS tags, and not hand crafted rules. For the weights, any weights which give a sufficiently high \chi_i’ value can be used to emphasize the retention importance. Therefore, we do not have different values for the non-unity weights. We set the value of the weights equal to 10 for our experiments. It is true that we are talking about the ROUGE scores for our experiments. However, it has been mentioned that the ability of our summaries to retain important words through weighted linkage generates better qualitative results. Examples can be seen in Figure A.4 of Appendix. Also, the agreement of 86.67% for our summary as the best summary is also partly due to the retention of important words.
>
> R3 : Motivation of telegraphic sentence compression with compression factor being 0.7 in comparison to 0.37 in Malireddy et. al
>
> As can be seen from the gold summaries for our test data, sentence compression with a compression factor less than 0.7 is prone to loss of important information. As stated in the Future Work section, we plan to utilize the sentence based telegraphic summarization as an input to conventional extractive summarization techniques so as to include more information in extractive summaries as a possible application of the current work. However, the true motivation of sentence level based summarization comes from the intuition that in cases like fictional stories, every sentence might be important and needs to be retained in the summary, as has been explained in the Introduction section. The work of Malireddy et al. was a work stating the utility of telegraphic summarization. We do not intend to achieve a compression factor they have achieved in the dataset, owing to the lack of context at sentence level. We agree, nevertheless, that the cognitive load of reading telegraphic sentences being less than actual sentences is debatable.
>
> R3 : Minor comments regarding mistakes in diagrams
> Thanks. Will make needed changes.

---

### Official Review · AnonReviewer1 · 2018-11-03
**Reasonable approach (but somewhat unsurprising) for a not entirely convincing problem**

**Rating:** 4
**Confidence:** 4

**Review:**

The authors introduce the problem of telegraphic summarization: given a sentence, we want to reduce its size while retaining its meaning, with no penalty for grammatical mistakes. The main application presented by the author is that of summarizing fictional stories and plays.

The setting proposed by the author prescribes that the summarized sentence can be obtained by the input sentence by dropping some words. So, for example, the simplest baseline for this problem would consist of simply dropping stop words.

The approach proposed is basically an auto-encoder, consisting of a 2-step encoder-decoder network: in the first step, the sentence is encoded into a vector which is in turn decoded to a (smooth) indicator vector to mask words in the sentence; in the second step, the masked sentence is encoded into a vector, which is in turn decoded into the output (summarized) sentence.

The optimization is a tradeoff between recoverability of the input sentence and norm of the indicator vector (how many words are dropped). In order for the network not to learn repetitive masking patterns (eg, drop first half of the sentence, or drop every other word), an additional loss is introduced, that penalizes keeping easily inferable words or dropping hard-to-infer words.

Concerns:
- the problem doesn't seem to be well-motivated. Also, the length of the obtained summarized sentences is ~70% that of the original sentences, which makes the summaries seem not very useful.
- the proposed complex architecture seems not to justify the goal, especially considering that simply dropping stop words works already quite well.
- In order for the presented architecture to beat the simple stop-words baseline, an additional loss (L4, linkage loss) with "retention weights" which need to be tuned manually (as hyper-parameters) is required.
- there's not enough discussion about the related work by Malireddy et al, which is extremely similar to this paper. A good part of that work overlaps with this paper.
- comparison with literature about abstractive summarization is completely missing.

Minor comments:
- Figure 1: Indicator Encoder should be Indicator Decoder.
- Are negations part of your stop words? From your discussion, you should make sure that "not", "don't", "doesn't", ... do not belong to your stop word set.
- How did you optimize the hyper-parameters r (desired compression), the regularization weights, and the retention weights?
- Were pre-trained word embeddings used as initialization?
- What's the average compression of golden sentences?

---

> ### Author Response · Authors · 2018-11-25
> **Response to Reviewer 1**
>
> R1 : The problem is not well-motivated and summary lengths and compression ratio around ~0.7
>
> Although, we understand the intuition that the problem will be better motivated if run on entire documents, sentence level telegraphic summarization is also important as there might be cases where each sentence of a text is important in some way. In these cases information is not repeated multiple times, rather a single piece of information can be conveyed across multiple sentences. To perform telegraphic style summarization for these cases, we either need to do a sentence-by-sentence telegraphic summarization or introduce context along with the input sentence as summarization. In this work, we chose to explore the possibility of single sentence telegraphic summarization with addition of context as a possible future work.
>
> Regarding the compression ratio, the compression of any sentence is much more difficult    and therefore cannot be compared with the compression ratio for documents. This is also reinforced by the gold corpus dataset we have created where the compression factor is just above 0.7.
>
> Compression factor varies according to input sentence len?
>
> R1 : The complex network doesn’t justify the goal, considering dropping stop words well
>
> The proposed network works well across sentence length, which is not achieved using stop word removal. Additionally, the network does not blindly drop some words (including those in the stop list) to compress the sentence. Instead, the retention/dropping of each word is decided on the basis of its inferebility and importance. This is a clear advantage of the proposed network over stop word removal.
>
> R1 : Addition of linkage loss and addition/fine-tuning of retention weights
>
> The linkage loss is added so as to suppress the masking effect in the summaries. This was a problem we faced independent of stop word retention/removal. Additionally, we do not fine-tune or use a handcrafted approach for deciding the retention weights. The retention weights of all word is set to be equal to 1, which is effectively not changing the input vector. The retention weight corresponding to all negation and subject words (these words were picked using their POS tags and not manually) is set to a high value (in our case 10), so as to simulate their importance in a sentence. Therefore, this is not handcrafted and is neither exclusively used to just outperform the stop words based baseline algorithms.
>
> R1 : Comparison with Malireddy et al.
>
> The work by Malireddy et al. was just used to motivate and demonstrate the usefulness of telegraphic summarization. Apart from this, there is no overlap between the paper and the proposed work. We propose an architecture for telegraphically summarizing a sentence while the authors in Malireddy et al. propose a dataset of documents containing fictional stories which are summarized telegraphically.
>
> R1 : Comparison with abstractive summarization
>
> The proposed work is concerned with telegraphic summarization, which is a type of extractive summarization as had been mentioned in the paper. Therefore, it is not possible to compare our approach with abstractive summarization techniques. The test dataset, that we have generated, follow the guidelines such that they only contain words present in the original sentence, rendering a comparison with abstractive techniques infeasible, as it will naturally lead to incorrect ROUGE scores.
>
> R1 : Minor comments
>
> We thank R1 for carefully going through the manuscript and pointing out the typos. We will correct these errors in the later version of the manuscript. Regarding stop words, these words are simply the most frequent 5% of the words. We have not made any further attempts to filter out words from this list. The hyperparameters were optimized empirically by analyzing the individual loss function behaviours. We have not set the retention weights individually for each of the words. The retention weights for all words being negation or subject words has been set 10, while for the other words the retention weight is 1. Pre-trained word embedding were not used for initialization, which is why no such mention has been made in the draft. Also, as mentioned in the draft and in response to a previous concern, the average compression length for the gold summaries is equal to 0.702. We will make these points clearer in the next version of the manuscript.

---

### Meta-Review · Area_Chair1 · 2018-12-14

**Confidence:** 4
**Recommendation:** Reject

**Metareview:**

This paper presents methods for telegraphic summarization, a task that generates extremely short summaries.

There are concerns about the utility of the task in general, and also the novelty of the modeling framework.

There is overall consensus between reviewers regarding the paper's assessment the feedback is lukewarm.